# Therapeutic Application of Modulators of Endogenous Cannabinoid System in Parkinson’s Disease

**DOI:** 10.3390/ijms25158520

**Published:** 2024-08-05

**Authors:** Leonid G. Khaspekov, Sergey N. Illarioshkin

**Affiliations:** Brain Science Institute, Research Center of Neurology, Volokolamskoye Road, 80, 125367 Moscow, Russia

**Keywords:** endogenous cannabinoid system, Parkinson’s disease, motor and non-motor symptoms, cannabis therapeutic application

## Abstract

The endogenous cannabinoid system (ECS) of the brain plays an important role in the molecular pathogenesis of Parkinson’s disease (PD). It is involved in the formation of numerous clinical manifestations of the disease by regulating the level of endogenous cannabinoids and changing the activation of cannabinoid receptors (CBRs). Therefore, ECS modulation with new drugs specifically designed for this purpose may be a promising strategy in the treatment of PD. However, fine regulation of the ECS is quite a complex task due to the functional diversity of CBRs in the basal ganglia and other parts of the central nervous system. In this review, the effects of ECS modulators in various experimental models of PD in vivo and in vitro, as well as in patients with PD, are analyzed. Prospects for the development of new cannabinoid drugs for the treatment of motor and non-motor symptoms in PD are presented.

## 1. Introduction

Cannabis is a genus of herbaceous plants in the cannabis family (Cannabaceae). The history of the therapeutic use of cannabis goes back a long way. Five thousand years ago in China, it was used to treat malaria, constipation, and rheumatic pains; mixed with wine, cannabis served as an analgesic for surgical manipulations. The therapeutic effects of cannabis as an anticonvulsant, muscle relaxant, antispasmodic, and hypnotic remedy are known [1].

The use of cannabis for medical purposes began to rapidly expand in the first half of the 19th century, when drugs based on it became “over-the-counter” in England, and in 1854 they were included in the US Pharmacopeia. Cannabis has been recognized as an effective drug in senile insomnia, neuralgia, migraines, and gouty pains epileptiform seizures, and, with careful use, cannabis was considered one of the most valuable medicines [1].

At the beginning of the 20th century, the use of cannabis in medicine declined dramatically due to the variability in the activity of herbal medicines, their instability during storage, the appearance of unpredictable effects when ingested, the emergence of effective alternative drugs of synthetic origin, commercial pressure, and concerns about the use of cannabis as a drug. In 1928, after the ratification of the Geneva Convention and other legislative acts, cannabis was outlawed in many countries.

More than 400 chemicals have been identified in cannabis, but its main active ingredients are cannabinoids, substances that are classified as aryl-substituted meroterpenes. 

Cannabinoids are found in the stems, leaves, flowers, and seeds of cannabis, as well as in the resin secreted by female plants. More than 60 cannabinoids are known, but the most studied are cannabinol (CBN), cannabidiol (CBD), and the main psychoactive component of cannabis, Δ-9-tetrahydrocannabinol (THC) [2]. CBN and CBD are additive, synergistic, or antagonistic compounds to the effects of THC. The cellular targets of THC are cannabinoid receptor (CBR) types 1 and 2 (CBR1 and CBR2) [2]. Their molecular structures are available and presented elsewhere [3,4].

Intensive research on the properties of natural cannabinoids in parallel with the development of their synthetic compounds with high activity and stereoselectivity has revealed the main physiological functions modulated by this class of substances. The discovery of CBRs and the development of highly selective powerful cannabimimetics has contributed to the identification of a family of lipid transmitters serving as endogenous ligands of CBRs—endocannabinoids and arachidonic acid derivatives. The main endocannabinoids include arachidonoylethanolamide (arachidonoylethanolamide/AEA), or anandamide (from Sanskrit Ananda, bliss), which was discovered in 1992, and 2-arachidonoylglycerol (2-arachidonoylglycerol/2-AG), identified in 1995. It was found that endocannabinoids are very similar to synthetic cannabimimetics in their physiological properties. In addition to the major phytocannabinoids, cannabis produces over 120 other cannabinoids that are referred to as minor and/or rare cannabinoids [5]. These cannabinoids are produced in smaller amounts in the plant and act as agonists and antagonists at multiple targets including CBR1 and CBR2, transient receptor potential channels, peroxisome proliferator-activated receptors and others [5]. The subsequent description of the complex biochemical pathways of synthesis, release, transport, and degradation of endocannabinoids has contributed to the formation of the idea of a new universal lipid signaling system, which was named the endogenous cannabinoid system (ECS) [2].

ECS arose in the early stages of evolution and performs important regulatory functions in all vertebrates. According to the latest results of physiological, pharmacological, and morphological studies, the main neurophysiological effect of cannabinoids is retrograde regulation of neurotransmitter release by activating presynaptic CBRs located at the axonal terminals. CBRs belong to the family of G-protein-related receptors and are widely represented in many structures of the brain, such as the basal ganglia (including the globus pallidus and the substantia nigra), limbic and paralimbic systems, cerebellum, and cerebral cortex, whose functions are associated with the control of motor activity, cognition, emotional reactions, motivation behavior, and homeostasis (Figure 1). CBR1 is found predominantly on GABAergic, glutamatergic, and monoaminergic neurons presynaptically, while CBR2 is expressed in peripheral immune cells, microglia, and neurons in the brain stem, striatum, and hippocampus [6,7].

Endocannabinoids are released “on demand” through receptor-mediated cleavage of membrane lipid precursors and serve as retrograde signaling messengers in GABA and glutamatergic synapses, as well as modulators of postsynaptic neurotransmission, interacting with other neurotransmitters. Endocannabinoids are transported retrogradely into the presynaptic terminal by a specific capture system and inactivated with the participation of two well-studied enzymes—fatty acid amide hydrolase (FAAH) and monoacylglycerol lipase (MAGL)—which are localized presynaptically and are mainly co-localized with GBR1. However, the detailed mechanisms of how the cannabinoid uptake system works remain unclear. The existence of the uptake system for endocannabinoids is confirmed by the effects of the blockers of anandamide transport, which induce anxiolytic effects at doses that do not interfere with motor activity [8]. Interestingly, several preclinical and clinical studies have demonstrated that CBRs play a crucial role in dopaminergic transmission in the basal ganglia circuitry, and the interplay between the ECS and dopamine system is important for modulating motor and non-motor clinical manifestations of Parkinson’s disease (PD) [7,9].

Postsynaptic depolarization causes Ca^2+^ entry through potential-dependent Ca^2+^ channels [10]. An increase in the intracellular Ca^2+^ concentration leads to the synthesis of endocannabinoids in the postsynaptic neuron. Endocannabinoid synthesis can also be stimulated by the activation of presynaptic metabotropic receptors, in particular, glutamate (mGlu1/5) and muscarinic (M1/M3) ones. Endocannabinoids are extruded from postsynapse and activate retrogradely presynaptic CBR1s, which, in turn, inactivate Ca^2+^ and activate K^+^ channels through the G-protein, leading to the suppression of neurotransmitter secretion [10].

Recent advances in pharmacology have made it possible to synthesize a number of compounds that target various components of the ECS (CBR agonists and antagonists, anandamide capture blockers, etc.) and are potent selective inhibitors of endocannabinoid degradation. This allowed us to investigate the physiological role of endocannabinoids and opened new strategies in the treatment of patients with numerous pathological conditions, including neurological diseases, mental disorders, and pain [11].

PD is the second most common neurodegenerative disease (100–250 per 100,000 population), occurring almost everywhere. The number of PD cases increases significantly in older age groups. Thus, in the group over 60 years of age, the disease occurs in 1% of people, and after 75 years of age it occurs with a frequency of up to 3–4% [12,13]. Existing approaches to the treatment of PD are aimed at compensating for dopaminergic deficiency in the nigrostriatal system (levodopa, dopamine receptor agonists, and MAO B inhibitors) and correcting the imbalance of other neurotransmitter systems of the central nervous system (amantadines, etc.) [14,15,16]. Although they lead to a clear symptomatic effect, they are not able to slow down the progression of the neurodegenerative process [17]. Moreover, long-term antiparkinsonian therapy leads to multiple side effects, which dramatically reduces the effectiveness of treatment and quality of life in patients with advanced-stage PD. Surgical approaches may be helpful in relieving different symptoms (such as tremor) but not in halting the course of the disease [16,18,19]. Thus, new safe and effective drugs are needed that provide both persistent symptomatic and disease-modifying (neuroprotective) effects in patients with PD [20].

It should be stressed that PD is characterized not only by well-known motor clinical manifestations (hypokinesia, rest tremor, muscle rigidity, and postural problems) but also by a variety of non-motor symptoms, such as autonomic dysfunction (constipation, orthostatic hypotension, hypersalivation, sweating, etc.), various sleep disorders, depression, apathy, cognitive decline, impaired sense of smell, pain, and others [21,22]. In patients with advanced-stage PD, non-motor symptoms may even have a greater impact on quality of life than motor disorders [23]. Moreover, traditional antiparkinsonian medications (levodopa, etc.) are usually ineffective against non-motor symptoms [16,17]. This is where the potential of new cannabinoid-based drugs may be in demand (Figure 1). It is interesting in this regard that polymorphism in the *CBR1* gene may affect the risk of one of the key non-motor manifestations of PD, depression [24].

## 2. The Use of ECS Modulators in Experimental Models of PD In Vivo and In Vitro

It has been shown that the ECS constituents are actively expressed in the basal ganglia, interacting with glutamatergic, GABAergic, and dopaminergic neurotransmitter systems, which suggests a high therapeutic potential of ECS modulators in PD [25,26]. CBRs are of special pathogenetic significance in PD, since they are localized on the striatum neurons together with the dopamine D1/D2 receptors [27,28].

Elevated AEA levels in the striatum were found in rodents with a 6-hydroxydopamine (6-OHDA) model of PD [29]. Increasing the level of AEA by inhibiting the enzyme of its hydrolysis (FAAH) prevented the death of dopaminergic neurons induced by the neurotoxin 1-methyl-4-phenyl-1,2,3,6-tetrahydropyridine (MPTP) and prevented the development of parkinsonian motor disorders in animals [30]. Elevated AEA levels, induced by the inhibition of FAAH, activated dopamine synthesis and attenuated the severity of dyskinesia by activating CBR1 in rats with a PD model [31]. Similarly, chronic inhibition of the 2-AG hydrolysis enzyme (MAGL) increased the 2-AG level, preventing motor disorders and protecting the nigrostriatal pathway from damage in mice with an MPTP-induced PD model [32].

Experimental studies of the effects of natural cannabinoids in the model of neurodegeneration have shown that THC prevents inflammation-induced cognitive damage [33]. The positive effects of THC were simulated by activation of CBR1s by their selective agonist, arachidonyl-2′-chloroethylamide (ACEA) [33]. Δ9-tetrahydrocannabivarin (Δ9-THCV), a phytocannabinoid with antioxidant properties, reduced 6-OHDA-induced motor deficiency and the loss of dopaminergic neurons in the compact part of the substantia nigra of rats and mice with PD models [34]. Similarly, in the rotenone PD model in rats, the phytocannabinoid β-caryophyllene prevented the gliosis, oxidative stress, and death of nigrostriate dopaminergic neurons induced by the neurotoxin [35], and VCE-003.2, an aminoquinone derivative of the non-psychotropic phytocannabinoid cannabigerol, reduced the intensity of the inflammatory process in brain tissue and improved behavioral characteristics in preclinical animal testing [36]. In a preclinical model of PD in rats, CBD had an antinociceptive effect, while the CBR1 agonist (AM251) prevented this effect [37]. In striate microglial cells of animals with experimental models of PD, CBR2 expression was significantly increased compared to the control [38].

Transgenic models of PD expressing mutations in the ‘parkinsonian’ genes, such as leucine-rich repeat kinase 2 (*LRRK2*), PTEN-induced putative kinase 1 (*PINK1*), α-synuclein, and others, are an important research tool and help to elucidate the role of the ECS in complex molecular pathways leading to neurodegeneration. In *LRRK2* transgenic mice expressing the G2019S mutation, the selective activation of the CBR2 partially reversed the deficits in the hanging-wire test, which was accompanied by normalization of autophagy markers in the basal ganglia [39]. In the same genetic model of PD, the stimulation of dopamine D2 receptor by quinpirole reduced the spontaneous and evoked excitatory postsynaptic currents in striatal spiny projection neurons, and this effect was mediated by a phospholipase C-dependent release of endocannabinoid, with subsequent activation of presynaptic CBR1 and reduced release of glutamate [40]. Madeo et al. showed dopamine-dependent CBR1 dysfunction at corticostriatal synapses in homozygous *PINK1* knockout mice [41]. It has been shown that PD involves an early aggregation of α-synuclein (αSyn) in the enteric nervous system [42]. Thy1-αSyn mice, which overexpress human αSyn, showed a lower density of enteric dopaminergic neurons compared with non-transgenic animals, and this decrease was prevented by neuroprotective effects of endocannabinoid-like mediators following the administration of dietary docosahexaenoic acid [43].

Since mitochondrial dysfunction and oxidative stress are believed to play a key role in the pathogenesis of PD and other chronic neurodegenerative diseases, antioxidant phytocannabinoids may be regarded as promising neuroprotective compounds [44,45,46,47,48,49].

Haghparast et al. studied the effect of marijuana on cognitive impairment caused by 6-OHDA (assessed in tests with the Morris water maze and the recognition of new objects) and on the expression of dopamine and cannabinoid receptors in the rat hippocampus (assessed using real-time PCR) [50]. It was found that marijuana improved spatial learning, prevented memory disorders caused by 6-OHDA, and restored the hippocampal levels of dopamine and CBR mRNA. In one of the papers [51], essential therapeutic mixtures of cannabis components were proposed to be suitable for screening medicinal compounds in PD models. A reductionist approach has been applied to determine the minimum necessary mixtures of these components that are amenable to pharmacological formulation. Screening of sixty-three variations of the original cannabinoid mixtures revealed the five most effective mixtures, which proved to be particularly attractive for therapeutic use. The results of this work have shown the importance of a reductionist approach for the development of therapeutic antiparkinsonian agents based on a mixture of cannabis with a controlled ratio of components.

The results of an active search for evidence of the therapeutic effect of phytocanabinoids on preclinical models of PD in vivo were discussed in a systematic literature review [52]. It has been shown that in most studies on rodents with PD models, the use of phytocanabinoids leads to a significant improvement in motor function and reduces the loss of dopaminergic neurons, with the involvement of antioxidant, anti-inflammatory, and anti-apoptotic mechanisms. The high neuroprotective potential of cannabinoids in different in vivo PD models is emphasized, which creates a basis for clinical trials and the therapeutic use of cannabinoids as preventive agents to reduce the risk of developing PD [52].

A recent paper by other authors [53] presents the results of a search for controlled comparative studies that evaluated the effect of cannabinoids or blockers of their transport on behavioral tests in animal models of PD. The data from this meta-analysis indicate the relief of motor symptoms and, thus, the feasibility of cannabis testing in a clinical setting.

In recent years, a number of in vitro and in vivo experiments related to the use of cannabinoids in PD and other chronic neurodegenerative diseases have been conducted, with CBD being mostly frequently used [54,55,56,57,58,59]. This cannabinoid does not have a pronounced psychotropic effect and its proportion in cannabis plant extracts can reach 40% [60]. In one of experimental preclinical studies, the effect of CBD on nociceptive reactions of mice with the 6-OHDA-induced PD model was studied: under the CBD exposure, the nociceptive pain threshold in these animals increased significantly [37]. The 3-hydroxyquinone derivative of CBD, VCE-004.8, prevented the death of tyrosine hydroxylase (TH)-positive neurons in the substantia nigra in mice with the 6-OHDA model of PD, in parallel with positive changes in the reactivity of astro- and microglia [61]. In the same work, the cytoprotective effect of VCE-004.8 was confirmed with in vitro experiments on SH-SY5Y neuroblastoma cell cultures exposed to 6-OHDA. Cell survival analysis showed that this effect of VCE-004.8 is mediated mainly by peroxisome proliferator-activated receptors but not CBR2, since it was eliminated by an antagonist of the former (T0070907) but not the latter (SR144528). In another study on the same cell line with the MPTP-induced model of PD [62], it was shown that CBD interferes with apoptosis by reducing Bax and caspase 3 levels, as well as the content of poly-ADP-ribose polymerase 1 (PARP-1) in the nucleus. The authors believe that the protective effects of CBD may be mediated by the activation of the AKT/mTOR pathway, since the mTOR inhibitor rapamycin eliminated the protective effects of CBD.

Recently, the results of a comprehensive study of the CBD effects on behavioral and biochemical parameters in mice with the MPTP-induced PD model have been published [63]. The authors showed that CBD preserved cognitive function and spontaneous movements; increased the levels of 5-hydroxytryptamine, dopamine, and interleukin (IL)-10 in the brain tissue (which was accompanied by a decrease in tumor necrosis factor-α, IL-1β, and IL-6); enhanced the expression of TH and B-cell lymphoma 2 (Bcl-2); reduced the levels of Bax and caspase-3; and decreased the expression of the inflammasome pathway mediated by caspase-1/IL-1β.

In experiments with a 6-OHDA model of PD, the ability of CBD to inhibit the activity of glycogen synthase-3β kinase, the main inhibitor of the WNT/β-catenin signaling pathway controlling oxidative stress and inflammation, was shown [64,65]. In experiments on transgenic mice with a PD model, it was shown that CBD exposure led to a reduction in motor deficiency and an improvement in motor coordination in a modified forced swimming test, as well as an improvement in the biochemical parameters of the biosynthesis of fatty acids, arginine, β-alanine, and pantothenate/CoA [66]. In in vitro experiments, CBD showed a neuroprotective effect on MPTP toxicity, restoring the expression of axonal and synaptic proteins and reducing microglial activation [67].

Selective CBR2 agonists attenuate the degeneration of dopamine neurons and axonal terminals in rotenone-induced, MPTP-induced, and 6-OHDA-induced models of PD through their anti-inflammatory and antioxidant activities [38,68,69,70]. Therefore, the use of CBR2 ligands is currently regarded as a new promising therapeutic approach in neuroinflammatory disorders, such as PD, Alzheimer’s disease, and multiple sclerosis [71]. The CBR1 has an anti-inflammatory role and can interact with the angiotensin type 2 receptor, which in the central nervous system is involved in mechanisms related to neurodegenerative and neuroinflammatory diseases [72]. Moreover, some cannabinoids have recently been demonstrated as robust antioxidants that might protect the nerve cells from degeneration even when CBRs are not triggered [73]. As cannabinoids are thought to have properties to slow or presumably cease the steady deterioration of the brain’s dopaminergic systems, their use in combination with currently available drugs has the potential to introduce a radically new paradigm for treatment of PD [73].

## 3. Modulators of ECS in Clinical Studies in Patients with PD

The results of preclinical and clinical studies have shown that CBR1s in PD modulate motor symptoms and the activity of cognitive information processing systems [74,75]. In neurons and astrocytes of the substantia nigra of patients with PD, the expression of CBR2 was significantly increased compared to the control [38,76]. An increase in the level of AEA in the cerebrospinal fluid of patients with PD was found [77,78], indicating a compensatory protective role of AEA. According to the results of positron emission tomography (PET) [79,80], significant regional changes in CBR1 levels were found in patients with PD, unrelated to the severity of levodopa-induced dyskinesia. In PD, low CBR1 levels in the middle superior frontal gyrus were associated with general cognitive disfunction, impaired executive functions, and poor episodic memory, and in patients with severe visual-spatial dysfunction, CBR1 content decreased in the precuneus, motor cortex, middle cortex, additional motor cortex, lower orbitofrontal gyrus, and thalamus [75]. Synthetic nonselective agonists of CBR, HU-210, and WIN55,212-2 protected nigrostriatal neurons probably via an antioxidant mechanism secondary to CBR2 activation [81].

An open pilot study showed that CBD in combination with standard antiparkinsonian therapy minimized the risk of psychotic symptoms in patients with advanced-stage PD, not affecting cognitive or motor symptoms [82]. In another double-blind randomized clinical trial involving 21 patients, CBD improved quality of life indicators over 6 weeks of treatment due to its effect on non-motor symptoms of the disease, while the motor functions of patients did not change [83]. Similarly, a pilot study involving four PD patients with a rapid eye movement sleep behavior disorder showed a decrease in the frequency of agitation, kicks, and nightmares after the treatment course based on 99.9% purified CBD [84].

The results of an open observational study involving 22 patients with PD demonstrated a significant decrease in the severity of resting tremor, muscle rigidity, and bradykinesia, as well as of non-motor symptoms of the disease (sleep disorders, pain syndrome, etc.) after smoking marijuana [85]. On self-assessment via the internet, 454 identified PD patients who consumed cannabis daily for more than 12 months reported lower disability and fatigue, and almost half of the patients reported a decrease in the consumption of prescribed antiparkinsonian drugs since the beginning of the cannabis use [86].

Another study conducted a telephone survey of 47 PD patients who used cannabis in the form of smoking marijuana for an average of 19 months (in Israel, such marijuana use is allowed under strict medical supervision). It turned out that most patients experienced improvements in falls, pain, depression, tremor, muscle rigidity, and sleep at the beginning of the drug use [87]. Cannabis extracts improved motor activity and relieved pain symptoms in patients with PD and had a dissociative effect on the pain threshold when exposed to high or low temperatures, indicating that peripheral and central nociceptive pathways can be modulated by cannabinoids [88].

One of the most serious problems in PD is the development of drug dyskinesia, a typical and difficult-to-treat complication of long-term levodopa therapy. There have been several reports on the effectiveness of cannabis in levodopa-induced dyskinesia in patients with PD [89,90]. However, in randomized, double-blinded studies, the antidyskinetic effect of cannabis in PD has not been confirmed [91,92]. It can be concluded that the results obtained to date regarding the antidyskinetic properties of cannabinoids and their action on other motor manifestations of the disease (tremor, etc.) remain insufficient to judge their effectiveness in accordance with these indications.

The potential benefit of cannabinoids in PD was confirmed by the results of a randomized, double-blinded, placebo-controlled study of a synthetic analog of THC, nabilone (NMS-Nab) [93], which included 47 patients with PD who had stable motor manifestations of the disease and pronounced, disabling non-motor manifestations (≥4 points on a non-motor scale MDS-UPDRS-I). At week 4 of the study, the average change in MDS-UPDRS-I was 2.63 (95% confidence interval, 1.53–3.74, *p* = 0.002) in the placebo group compared with 1.00 (−0.16–2.16, *p* = 0.280) in the nabilone group (difference: 1.63, *p* = 0.030). In other terms, the rate of increase in non-motor disorders in the placebo group was significantly higher than in the nabilone group. There were no serious adverse events in the nabilone group. The resulting significant effect of nabilone on non-motor symptoms in patients with PD was primarily due to its positive effect on anxiety and sleep problems at night. In 77.4% of patients with PD associated with sleep disorders, nabilone was effective in reducing different domains of dyssomnia [94].

The use of a tincture containing THC and CBD in a 1:1 ratio led to the relief of different symptoms such as seizures/dystonia, pain, spasticity, lack of appetite, dyskinesia, and tremor in patients with PD and significantly reduced concomitant use of opioid drugs [95]. An analysis of 13 papers on the use of cannabinoids in PD showed that CBD, THC, and nabilone (a derivative of THC) dose-dependently and stably alleviated motor symptoms, and, in addition, THC reduced pain intensity and CBD improved the psychiatric condition of patients [96].

Yenilmez et al. studied the attitude of patients to the use of medical cannabis (MC) in the treatment of PD in Germany [97]. Using a survey based on a questionnaire, the authors assessed general knowledge of and interest in MC, the frequency and methods of its use, its effectiveness, and tolerability. In total, 1348 questionnaires were analyzed. As many as 51% of the participants were aware of the legality of the use of MC, 28% of the different routes of administration, and 9% of the difference between THC and CBD. Cannabis use was reported by 8.4% of patients with PD, which was associated with a younger age, living in large cities, and better knowledge of the legal and clinical aspects of PD. More than 40% of cannabis users reported reduced pain and muscle spasms. Stiffness/akinesia, freezing, tremor, depression, anxiety, and restless legs syndrome subjectively improved by more than 20%, and the overall tolerability of cannabis was good. Symptom relief was reported by 54% of participants who used CBD orally and by 68% of those who inhaled cannabis fumes containing THC. Compared with CBD intake, inhaling THC more often reduced akinesia and stiffness (50.0% vs. 35.4%; *p* < 0.05). In total, 65% of respondents reported their interest in using MC. Thus, MC was considered by many patients with PD as an appropriate therapeutic option.

The results of cannabis treatment of patients with PD were assessed in a meta-analysis of five randomized controlled and eighteen non-randomized trials [98]. In most patients, favorable results were noted regarding the relief of tremor, anxiety, pain, and improved quality of sleep and life in general. The authors emphasized the need for further well-planned randomized trials.

Noteworthy are the results of a randomized, double-blinded, placebo-controlled cross-examination of the effect of CBD on anxiety and parameters of tremor (frequency and amplitude) in the fear of public speaking test in 24 patients with PD [99], with an additional assessment of the heart rate and systemic blood pressure. It was found that acute administration of CBD weakened the anxiety and reduced the amplitude of tremor in this anxiogenic situation.

## 4. Conclusions

The above indicates the undoubted therapeutic potential of the modulation of the ECS in PD [100,101]. In recent decades, the ECS has attracted considerable interest as a potential therapeutic target for numerous disorders of the nervous system. Since PD is, clinically, a very polymorphic condition with a variety of motor and non-motor manifestations, it is a useful kind of “model” for assessing the multidimensional action of ECS modulators and is an adequate object for studying the cellular and molecular mechanisms of their action. Cannabinoids and endocannabinoids hold promise as disease modifiers for the prevention or treatment of neurodegenerative diseases [49]. Experimental and clinical experiences of using ECS modulators in PD and other neurodegenerative diseases create a basis for further intensive therapeutic studies of cannabis and its derivatives in chronic neurodegeneration. It is necessary to improve complex experimental protocols for ECS testing, increase the reliability of animal models of PD, and further optimize the processes of preparing the cannabis drugs used in experiments. Before wider use of cannabis in clinical practice, it is necessary to clarify the most preferred cannabinoids suitable for patients with PD, their optimal dosage and routes of administration and frequency of use, as well as to determine the spectrum of symptoms (non-motor and motor) for which the use of cannabis may be most optimal.

This systematic review included publications without restrictions on the year of publication, selected using the following keywords: Parkinson’s disease, endogenous cannabinoid system, cannabinoids, cannabinoid receptors, cannabis, and therapeutic application. The following databases were used: MEDLINE, EMBASE, PubMed, and Web of Science core collection.

## Figures and Tables

**Figure 1 ijms-25-08520-f001:**
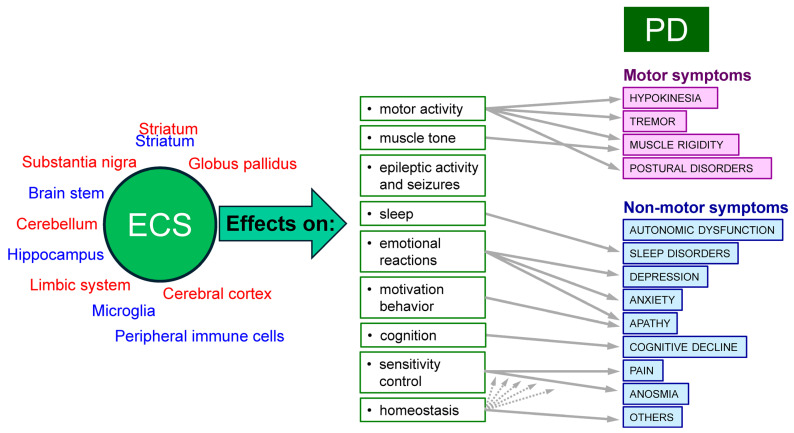
ECS and clinical manifestations of PD. The predominant localization of CBR1s (red) and CBR2s (blue) are shown on the left. In the center and on the right, the relationships between the biological effects of ECS and the clinical symptoms of PD are shown. Possible “clinical targets” for ECS modulators are indicated by arrows.

## Data Availability

No new data were created or analyzed in this study.

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
