# Peer review of "Therapeutic Application of Modulators of Endogenous Cannabinoid System in Parkinson’s Disease"

_ijms, 2024, doi:10.3390/ijms25158520_

Round 1

Reviewer 1 Report

Comments and Suggestions for Authors

The investigation in the therapeutic potential of pharmacologically modulating specific elements of the endocannabinoid system in Parkinson’s disease has been an important and biomedically relevant objective in the last 15-20 years. This research has generated numerous research articles and reviews based on the progressive advances in this objective. This submission is a new attempt to review the topic, but it adds poorly to previous and recent reviews that have addressed the same question. This is possibly the major defect of this manuscript together with an unclear and unstructured organization, as well as with some important studies that are missing in the text (e.g., those studies that have investigated the potential of activating or inhibiting the CB1 receptor in levodopa-induced dyskinesia) and studies that are not adequately presented. Other important deficiency is the lack of a detailed description of the location of endocannabinoid elements in the basal ganglia circuitry.

Specific comments:

1.     Abstract: the first sentence is not correct as the first demonstrations on the therapeutic potential of cannabinoids for Parkinson’s disease were published in 2005 and even earlier.

2.     Line 45: Cannabinol is a psychoactive cannabinoid as it binds to CB1 receptor with an affinity that is not so different to Δ9-THC. I am afraid that authors confused cannabinol with cannabigerol, which is non-psychoactive.

3.     Line 56: the name anandamide also considers the amide structure of AEA.

4.     Line 58: a few words on the so-called minor endocannabinoids would be necessary.

5.     Line 78: the existence of the uptake system for endocannabinoids has been a controversial issue. This should be mentioned, updated and discussed.

6.     Lines 78-80: authors should address with more detail the presynaptic or postsynaptic location of FAAH and MAGL enzymes, and to relate this with the location of the CB1 receptors.

7.     Line 102: the word “modern” is not a good option for describing a therapy that was initiated in the 60s of the past century.

8.     Lines 142-145: the study published in ref. 20 does not include the results that are described in the sentence.

9.     Lines 235-238: authors should comment on the cell substrates (e.g., glial cells, neurons) in which the changes of the CB2 receptor take place. There are some key articles on this question that are not mentioned.

10. The review will receive an important benefits in terms of clarity with the addition of several diagrams or schemes that summarize the major results and findings.

Comments on the Quality of English Language

See before

Author Response

Reviewer 1.

- We have added a large number (more than 20) of new references to important missing works on this issue, including works on the pathophysiological relationship of the ECS with clinical manifestations of Parkinson's disease, as well as on the location of endocannabinoid elements in the basal ganglia and other cerebral circuitries (see, for example, lines 71-77, 79, 80, 84, 85, 89-92, 134-138 and others).

- We corrected the first sentence in the abstract.

- We corrected the description of psychoactive properties of cannabinoid and Δ9-THC (lines 42-44).

- We added a short description of minor endocannabinoids (lines 57-61).

- We updated several sentences concerning the uptake system for endocannabinoids (lines 85-88).

- Synaptic location of FAAH and MAGL enzymes and their relation with the location of the CB1 receptors are indicated (lines 83-85).

- The word “modern” while describing antiparkinsonian therapy has been omitted from the text (replaced by “existing’, line 110).

- In accordance with reviewer's comment â„–8, we have changed the sentence and the reference (see line 149-152, ref. # 33).

- Several additional new references have been added to the text.

We believe that the above changes have made our review more complete, understandable and logical.

Reviewer 2 Report

Comments and Suggestions for Authors

The study presented here is a literature review focused on modulation of the brain's endogenous cannabinoid system (ECS) with new drugs as a possible pharmacotherapy for Parkinson's disease.

The weakness of the reviewed work concern as follows:

•           The first serious shortcoming of the paper is the lack of presentation of the methodology of the review conducted in accordance with the PRISMA 2020 statement (doi: https://doi.org/10.1136/bmj.n71). Which is the current standard for conducting review work and transparently report why the review was done, what the authors did, and what they found. Additionally, this assures the reader that there is no random selection of literature in the report presented. I strongly recommend the addition of a Materials and Methods chapter containing the following information: Data Selection, Database Search Criteria, Researched Databases.

•           The second major drawback of the article is the low number of citations of papers from the last five years. This review summarises many old research papers (older than five years, i.e. out of 80 bibliographic entries, only 37 papers were published after 2019). Which seems a very poor treatment of the reader. The topic has evolved dynamically in recent times and focusing on old articles is not the best approach to review it. Additionally, citation 59 cries out for completion.

•           Since a large number of abbreviations are used along the text, I recommend that all used abbreviations should be listed at the end of the main body of text before references. This will certainly provide the reader with a better understanding of the context of the issues discussed. In addition, this is a kind of standard in contemporary scientific papers.

•           IJSM, as the name suggests, is a journal dedicated to molecules. In my opinion, this aspect needs to be commented on in the article. In the article, the authors review the structures of CBRs such as CB1 and CB2. However, an overview of the availability of their molecular structures is missing (look at the papers discussing the structures of CB1 https://doi.org/10.1016/j.cell.2016.10.004 and CB2 https://doi.org/10.1016/j.cell.2018.12.011 ). I also believe that, relevant molecular studies of cannabinoid complexes with these proteins should be commented on. I also suggest using databases such as RCSB PDB (https://www.rcsb.org) or UniProtKB (https://www.uniprot.org/) for this purpose. Additionally, I propose to select the cited examples taking into account a specific criterion, e.g. highest resolution value or time of publication or others. Given the current relevance of the topic under discussion, there are many deposited structures in the databases and it does not make sense to cite them all.

•           For similar reasons as above, it is advisable for the authors to include a figure with structures of the compounds discussed in their review.

I recommend publication after major revision.

Author Response

Reviewer 2.

- We added the information on the methodology of the review conducted (Data Selection, Database Search Criteria, Researched Databases) as a separate paragraph at the end of the text (lines 360-364).

- We added a significant number of new citations (more than 20) of papers published in the last 3-4 years (see, for example, ref. # 5, 7, 9, 13-16, 19, 67-69, 98 and others).

- All used abbreviations have been listed at the end of the main body of text before references.

- We mentioned the availability of the molecular structures of CBR1 and CBR2, with the corresponding references (lines 45, 46). Since this review is devoted mainly to the physiological and clinical effects of ECS ligands in Parkinson's disease (in the experiment and clinic), it seems to us not entirely justified to discuss the molecular structures of CBRs in more detail.

We believe that the above changes have made our review more complete, understandable and logical.

Reviewer 3 Report

Comments and Suggestions for Authors

Presented manuscript is the review article. Authors aimed to present connection of modulators of endogenous cannabinoid system with therapy of Parkinson’s disease. In my opinion this review in present form is not suitable to publish in IJMS. Review article should order the data which can be found in literature. The subject of this article is interesting, but it seems like material gathered on this subject. Introduction allows to recognize what about the Authors would like to talk but further is not formulated which targets are taken into the consideration. Which mechanisms of action could be reasonable and why. Mentioned are CB1R and CB2R but it is not clear why. Several names of compounds (AM251, VCE-003.2, VCE-004.8, T0070907, SR144528…) were also mentioned but without of presenting them. Modulators of endogenous cannabinoid system should be systematized. It seems that Authors have used key words to gather literature on aimed subject and described which was considered in them. How was the material found? Which data bases were used? Which are the main conclusions coming from this article.

Additionally some type and grammar mistakes could be find in text e.g.:

-          Not finished sentence at page 5 line 25 and page 6 line 25

-          Page 2 line 58: sentence starting with: It was found that ,… should be rearranged

-          Page 2 line 71: strsuctures

-          Page 3 line 14 and page 4 line 14: hydroxydophamine

-          Page 5 line 22: not finished sentence starting with: “In experiments…

Comments on the Quality of English Language

English should be improved. Details included in Comments for Authors

Author Response

Reviewer 3.

- We have tried to present all the material in a more systematic manner throughout the text, including by adding and discussing a significant number of new publications on key aspects of the problem over the past 3-4 years (more than 20 such new references). We have also added data on the methods of collecting information and the databases used at the end of the main body of the text (lines 360-364).

- We corrected all the grammar and type mistakes indicated by the reviewer.

We believe that the above changes have made our review more complete, understandable and logical.

Round 2

Reviewer 3 Report

Comments and Suggestions for Authors

The manuscript was reworked and  reorganized. It is suitable to publish in IJMS (English should be checked

Author Response

Dear Editor:

Attached is a new version of our manuscript “Therapeutic Application of Modulators of Endogenous Cannabinoid System in Parkinson's Disease”, with appropriate changes made based on the editor's comments (minor points).

We made additional revisions as follows.

Editor's comments:

- Given the complexity of PD, diagrams or schemes that summarize the major results on ECS were not added and are missing (see also comment by reviewer 1, No 10). This addition will be instrumental for the understanding of the content of this review by a more general readership.

Response:

  • We have prepared a comprehensive figure illustrating the relationship between the endogenous cannabinoid system and Parkinson's disease. All the most important aspects of the review are reflected in it. The figure is attached to this letter (see separate file “Figure”), the legend to this figure is given at the end of the text file.

Editor's comments.

A discussion of relevant genetic models of PD such as leucine-rich repeat kinase 2 (LRRK2), alpha-synuclein, PINK1 and others in the context of ECS is not found and should be included: A) For LRRK2 at least these publications should be relevant for the discussion: Palomo-Garo C, Gómez-Gálvez Y, García C, Fernández-Ruiz J. Targeting the cannabinoid CB2 receptor to attenuate the progression of motor deficits in LRRK2-transgenic mice. Pharmacol Res. 2016 Aug;110:181-192. doi: 10.1016/j.phrs.2016.04.004. Epub 2016 Apr 6. PMID: 27063942. Tozzi A, Durante V, Bastioli G, Mazzocchetti P, Novello S, Mechelli A, Morari M, Costa C, Mancini A, Di Filippo M, Calabresi P. Dopamine D2 receptor activation potently inhibits striatal glutamatergic transmission in a G2019S LRRK2 genetic model of Parkinson's disease. Neurobiol Dis. 2018 Oct;118:1-8. doi: 10.1016/j.nbd.2018.06.008. Epub 2018 Jun 13. PMID: 29908325. B) For alpha-synuclein, PINK1 and others there may be a number of potentially relevant publications. Authors are requested to focus on papers including molecular aspects of the current topic.

Response:

  • We have added a large new paragraph to the text (see page 4) reflecting the role of transgenic models of PD in studying the endocannabinoid system in genetic forms of the disease. It discusses the main data from the articles indicated by the editor and some additional articles concerning the study of the endocannabinoid system in transgenic models expressing mutations in the LRRK2, PINK1 and alpha-synuclein genes. Accordingly, 5 new references have been added to the list of references, and the numbers of other references in the text have been changed. Also, two new abbreviations have been added to the list of abbreviations (page 8).

All changes are highlighted in yellow.

With best regards,

S.Illarioshkin, L.Khaspekov.